# The Effect of a Dominant Inhibitory p53 Protein on Stress Responses Induced by Toxic and Non-Toxic Concentrations of Anisomycin in PC12 Cells

**DOI:** 10.3390/biology14121634

**Published:** 2025-11-21

**Authors:** Renáta Schipp, Judit Varga, Judit Bátor, Mónika Vecsernyés, Zita Árvai, Petra Kele-Morvai, József Szeberényi, Marianna Pap

**Affiliations:** 1Department of Medical Biology, Medical School, University of Pécs, 7624 Pécs, Hungary; renata.schipp@aok.pte.hu (R.S.); judit.varga@aok.pte.hu (J.V.); judit.bator@aok.pte.hu (J.B.); monika.hengl@aok.pte.hu (M.V.); petra.morvai@aok.pte.hu (P.K.-M.); jozsef.szeberenyi@aok.pte.hu (J.S.); 2National Laboratory of Virology, University of Pécs, 7624 Pécs, Hungary; zita.arvai@aok.pte.hu

**Keywords:** anisomycin, apoptosis, p53 protein, PC12 cell line, stress signaling, exosomes

## Abstract

Anisomycin is an antibiotic derived from the bacteria Streptomyces griseolus. As a ribotoxin, anisomycin inhibits eukaryotic protein synthesis by damaging the 28S ribosomal RNA and inhibiting its peptidyl transferase activity. This triggers a cellular process called ribotoxic stress response (RSR). Depending on dosage, duration of treatment, and cell types, anisomycin can stimulate the main stress kinases p38 MAPK and JNK, and different signaling pathways leading to apoptosis (programmed cell death) or cell survival. In this study, we compare the signaling effects of toxic/protein synthesis-inhibiting (1 µg/mL) and non-toxic (10 ng/mL) concentrations of anisomycin in PC12 cells. We also used a PC12 cell line expressing a dominant inhibitory p53 protein to study the role of the p53 tumor suppressor protein in the signaling events caused by anisomycin. We found that in the presence of active p53, anisomycin induced strong stress responses leading to apoptosis and release of stress-related proteins such as TRAIL and PKR through exosomes. In contrast, when p53 was blocked by the dominant-negative mutant, these stress and apoptotic responses were reduced. Our findings may help to understand how cells regulate the signaling of death or survival under ribotoxic stress conditions.

## 1. Introduction

Several naturally occurring translation-interfering toxins, termed ribotoxins, effectively target the ribosomes through a process called “ribotoxic stress response” [1]. The ribotoxin anisomycin is one of the most potent protein synthesis inhibitors [2]. By binding to the peptidyl transferase center of the eukaryotic 28S rRNA, anisomycin suppresses the peptidyl transferase reaction, thereby blocking peptide bond formation [3]. Depending on dose, time, and cell type, anisomycin can stimulate the stress-activated protein kinases, p38 mitogen-activated protein kinase (p38 MAPK), c-Jun N-terminal kinase (JNK), and other signal transduction pathways leading to apoptosis or cell survival [4,5,6,7]. Besides stress-activated kinases, dsRNA-activated protein kinase (PKR) may also play a significant role in ribosomal stress (for reviews, see [6,8]). In response to cellular stress, PKR undergoes caspase-dependent cleavage and separates its regulatory domain from its kinase domain. The latter activates full-length PKR molecules by phosphorylation, which in turn phosphorylates the translation initiation factor eIF2α, leading to inhibition of protein synthesis and consequently apoptosis [9,10].

Stress kinase pathways are stimulated even by low concentrations of anisomycin that do not affect protein synthesis [11]. The mechanism of anisomycin-stimulated signaling events is not fully understood, nor is the role of the p53 tumor suppressor protein, a key regulator of apoptosis and cell cycle in these signal transduction processes [12]. The cellular level and activity of the p53 protein depend on the condition of the cell. In proliferating cells, p53 protein levels are low, but various stress conditions lead to increased expression of the p53 protein [13,14]. After its activation, p53 up- or downregulates the transcription of several genes involved in the regulation of cell cycle arrest, DNA repair, and specific cell death pathways [15,16,17]. Some of these genes encode secreted proteins that may be involved in the communication between cells [18]. p53 was also found to be able to regulate exosome formation [19]. Exosomes carrying regulatory molecules are important mediators of intercellular signaling [20,21]. Such signaling molecules include tumor-necrosis-factor-related apoptosis-inducing ligand (TRAIL), a death ligand with a unique ability to induce apoptosis in a variety of cancerous or transformed cells [22,23]. TRAIL is a type II transmembrane protein of the TNF superfamily that binds to its cognate death receptors (DR4, DR5), leading to the activation of the extrinsic apoptotic pathway [24]. Its proteolytic product acts as a secreted ligand, while the transmembrane isoform may be involved in direct cell-to-cell signaling [25]. Some studies showed that TRAIL can be isolated from the supernatant of cultured cells in a microvesicular form [26,27], indicating an exosomal paracrine mechanism for this death ligand.

Exosomes are 30–200 nm-sized vesicles generated by the late endosomal compartment. They are produced by a wide range of cells and contain cytoplasmic components, transmembrane proteins, mRNAs, miRNAs, and DNA fragments. They play an important role in cell–cell communications and in some physiological and pathological processes such as antigen presentation or the regulation of apoptosis [21,28]. Since we found earlier (see below) that TRAIL was released into the culture medium of anisomycin-stressed PC12 cells, we found it justified to analyze the possible involvement of exosomes in this paracrine signaling.

A useful cell culture system to study the cellular processes described above is the PC12 rat pheochromocytoma cell line (designated as wtPC12 cells throughout this paper) and its subclones. They are widely used model systems to study signaling pathways regulating neuronal differentiation [29], neurotoxicity [30], cell survival [31], and apoptosis [32]. The p143p53PC12 cell line used in this study is a PC12 subclone expressing a dominant inhibitory p53 protein. This p53 variant carries a missense mutation in its DNA-binding domain. If overexpressed, it interferes with the binding of the endogenous p53 protein to its cognate enhancer element by being incorporated into the tetrameric complex of this transcription factor. The expression of p53-regulated genes is inhibited in the p143p53PC12 cell line; this subclone is thus suitable to study the p53 dependence of cellular processes [33].

In PC12 cells, anisomycin can trigger both pro- and anti-apoptotic processes depending on its concentration [11]. Low doses of anisomycin activate cell survival, whereas high doses induce apoptosis. In some other cell types, low doses of anisomycin [34,35] sensitize the cells to TRAIL-induced apoptosis. In our previous studies, we found that the protein synthesis-inhibiting concentration of anisomycin stimulated apoptosis in wtPC12 and p143p53PC12 cells with different time kinetics and sensitivity. The p143p53PC12 subclone was more resistant to the ribotoxic stress-mediated apoptosis by anisomycin than wtPC12 cells. There were also differences in the activation of cellular signaling events like the phosphorylation of the stress kinases (p38 MAPK, JNK), p53 protein, and eukaryotic initiation factor 2α (eIF2α), as well as the proteolytic activation of caspases and PKR. We also found that the protein synthesis-inhibiting concentration of anisomycin stimulated the expression of TRAIL and its release into the culture medium [36].

The present study aimed to extend the analysis of anisomycin’s effects on PC12 cells. The experiments reported here were designed to compare the effects of a protein-synthesis-inhibiting (1 μg/mL) and a subinhibitory (10 ng/mL) concentration of anisomycin in wtPC12 and p143p53PC12 cells. Cellular stress responses (including the activation of stress kinase pathways, extrinsic and intrinsic apoptotic mechanisms), the formation of exosomes, and the p53-dependence of these effects of anisomycin treatment have been analyzed.

## 2. Materials and Methods

### 2.1. Cell Cultures

PC12 rat pheochromocytoma cell lines wtPC12 (a gift from G.M. Cooper), and p143p53PC12 [33] were cultured in Dulbecco’s Modified Eagle’s medium (DMEM; Sigma-Aldrich, Budapest, Hungary) containing 5% fetal bovine serum (FBS) and 10% heat-inactivated horse serum (Invitrogen, Carlsbad, CA, USA) at 37 °C and 5% CO_2_. For exosome isolation, cells were cultured in DMEM containing 0.5% heat-inactivated horse serum (referred to as 0.5% serum medium throughout this paper). Cells were treated with 1 µg/mL or 10 ng/mL anisomycin (Sigma-Aldrich, St. Louis, MO, USA) for 0, 2, 4, 8, 12, or 24 h.

### 2.2. Western Blotting

Western blot analysis was carried out according to the protocol of Santa Cruz Biotechnology (Dallas, TX, USA) as described earlier [37]. Briefly, 5 × 10^6^ cells were plated into 100-mm plates, and the next day they were treated with anisomycin as described above. At the end of the treatment periods, the cells were scraped off and lysed in a protein isolation buffer (RIPA: 1% NP40, 0.5% Na-deoxycholate, 0.1% SDS, 1× phosphate-buffered saline) containing protease (protease inhibitor cocktail) and phosphatase inhibitors (phosphatase inhibitor cocktail; all components were purchased from Sigma-Aldrich). The protein concentration of the samples was determined, and equal amounts of lysates (35 μg) were fractionated by 12% SDS-polyacrylamide gel electrophoresis. The proteins were transferred onto polyvinylidene difluoride membranes (Thermo Fisher Scientific, Waltham, MA, USA). The following antibodies (purchased from Cell Signaling Technology, Beverly, MA, USA) were used: eIF2α, P-eIF2α, extracellular signal-regulated kinase (ERK), P-ERK, cleaved caspase-3, cleaved caspase-9, caspase-8, JNK, P-JNK, p38MAPK, and P-p38MAPK. An antibody against PKR was obtained from BD Biosciences (San Jose, CA, USA), and an anti-TRAIL antibody was from Santa Cruz Technology (Dallas, TX, USA). Horseradish-peroxidase conjugated secondary anti-mouse or anti-rabbit antibodies were purchased from Cell Signaling Technology. The immune complexes were visualized with ECL reagent (Millipore Corporation, Billerica, MA, USA). The results were visualized using a G-box gel documentation system (Syngene, Cambridge, UK) or were developed with Amersham Hyperfilm^TM^ECL (GE Healthcare, Budapest, Hungary). Full-length, uncropped blots corresponding to the cropped images shown here are presented in the Appendix A.

### 2.3. Extraction of Exosomes from Cell Culture Media

The exosome isolation reagent was used according to the manufacturer’s instructions (Thermo Fisher Scientific, Budapest, Hungary). Briefly, 5 × 10^6^ cells were counted into 100-mm plates, and the next day, culture media were changed to 0.5% serum medium for 24 h, and cells were treated with anisomycin as described above. After treatment, culture media were collected and centrifuged at 2000× *g* for 30 min to remove cells and debris. The clarified media were transferred to new tubes and mixed with 0.2 volumes of the Total Exosome Isolation Reagent by vortexing until a homogenous solution was formed. The samples were incubated at 4 °C for 30 min and then centrifuged at 10,000×  *g* for 10 min. The supernatants were aspirated and discarded, and the exosome pellets were resuspended in PBS buffer (1.36 M NaCl, 2.7 mM KCl, 4.3 mM Na_2_HPO_4_ × 7 H_2_O, 1.4 mM KH_2_PO_4_). Isolated exosomes were stored at −80 °C in PBS until further analysis. Western blot analysis was performed using the isolated exosomes from 100-mm plates.

## 3. Results

### 3.1. Both Low and High Concentrations of Anisomycin Cause Activation of Stress Kinases in wtPC12 Cells and PC12 Cells Expressing a Dominant Negative p53 Protein

Anisomycin strongly activates the stress-activated protein kinases p38MAPK and JNK in mammalian cells [4,5,36]. Phosphorylation of p38MAPK and JNK was detected in both cell lines independently of the concentration of anisomycin, but the kinetics of phosphorylation looked different (Figure 1a). The treatment with 1 μg/mL anisomycin induced sustained phosphorylation in both cell lines, which started early after anisomycin treatment and lasted for at least 24 h. Both the subinhibitory and inhibitory concentrations of anisomycin stimulated p38MAPK phosphorylation in a sustained manner. The phosphorylation of p38MAPK was stronger in wtPC12 cells. In contrast, the phosphorylation of JNK showed biphasic kinetics after high-dose anisomycin treatment in wtPC12 cells: it reached its maximum after 2–4 h, then transiently decreased and was reactivated again after 24 h. wtPC12 cells treated with the subinhibitory concentration of the drug showed a similar pattern of JNK phosphorylation, but activation appeared slightly weaker, and its biphasic nature was even more pronounced. Both the basal level and the anisomycin-stimulated phosphorylation were higher in p143p53PC12 cells. Interestingly, low-dose anisomycin had a stronger activating effect. The phosphorylation of other members of the MAPK family, ERK1 and 2, was strongly inhibited in wtPC12 cells independently of anisomycin concentration, with slow recovery. In contrast, in p143p53PC12 cells, both concentrations of anisomycin caused ERK 1 and 2 activation; ERK activation was particularly robust after low-dose anisomycin treatment. The treatments hardly affected the levels of these MAPKs.

Low and high concentrations of anisomycin had different effects on PKR activation and the phosphorylation of eIF2α (Figure 1a). The proteolytic cleavage of PKR in wtPC12 cells treated with 1 μg/mL anisomycin was strong and sustained, but did not take place after 10 ng/mL anisomycin treatment. Anisomycin did not induce PKR cleavage in the p143p53PC12 cell line. Proteolytically activated PKR catalyzes the phosphorylation of the regulatory α-subunit of eIF2 on Ser51, which leads to the inhibition of translation. The phosphorylation kinetics of eIF2α were similar to those of PKR cleavage in wtPC12 cells upon high concentration anisomycin treatment. Treatment with 10 ng/mL anisomycin hardly affected eIF2α phosphorylation or eIF2α levels. 1 µg/mL anisomycin stimulated eIF2α phosphorylation in the p143p53PC12 cell line as well, but to a much smaller extent. The level of eIF2α did not change significantly. The subinhibitory concentration of anisomycin did not affect PKR cleavage and caused a transient decrease in the phosphorylation of eIF2α in the p143p53PC12 cell line.

### 3.2. p53-Dependence of Anisomycin-Induced Caspase Activation

To analyze the effect of subinhibitory and protein synthesis-inhibiting concentration of anisomycin on intrinsic and extrinsic apoptotic pathways, the activation of caspase-9, caspase-8, and caspase-3 was analyzed in both cell lines by using anti-cleaved caspase-9, anti-cleaved caspase-8, and anti-cleaved caspase-3 antibodies (Figure 2). Proteolytic cleavage of procaspase-9 and procaspase-8 activates the intrinsic and extrinsic apoptosis pathways, respectively. Both initiator caspases activate the main executioner caspase, caspase-3. The proteolytic activation of these three caspases by 1 µg/mL anisomycin was stronger and happened earlier in wtPC12 than in p143p53PC12 cells. The low concentration of anisomycin also caused the proteolytic cleavage of caspase-9 in both cell lines and the activation of caspase-3 in wtPC12 cells, but this effect was delayed and much weaker than that of the toxic anisomycin treatment. The subinhibitory concentration of anisomycin could not cause the activation of caspase-8 in either of the cell lines. It thus appears that activation of these key apoptosis-mediating enzymes in anisomycin-treated cells is highly dose- and p53 protein-dependent.

### 3.3. Potential Involvement of Exosomal Signaling in the Apoptosis of Anisomycin-Treated PC12 Cell Lines

Anisomycin was found to stimulate the intrinsic apoptotic pathway, as indicated by strong caspase-9 activation (see above). The activation of extrinsic apoptotic signaling takes place through the binding of death ligands to their death receptors, leading to the proteolytic cleavage and activation of initiator caspase-8. This activation was also detected in both cell lines, although with different intensities after 1 μg/mL anisomycin treatment. To identify the mechanism of this effect, we analyzed the secretion and level of TRAIL protein in exosomes and cell extracts (Figure 3a). The protein has a soluble (21 kDa) and a transmembrane (34 kDa) isoform [25]. The presence of membrane-bound TRAIL in the culture medium was increased in wtPC12 cells; the amount of transmembrane TRAIL started to rise 8 h after treatment with 1 μg/mL anisomycin and lasted for at least 24 h. The amount of this isoform in wtPC12 cells treated with a low concentration of anisomycin was even slightly higher. We could not observe the transmembrane isoform in the culture medium of p143p53PC12 cells. Anisomycin treatment did not affect the secretion of the soluble form of TRAIL in the two cell lines. The intracellular level of TRAIL was not significantly affected in wtPC12 cells upon anisomycin treatment; the mutant cell line contained lower amounts of TRAIL protein as compared to wtPC12 cells.

Since PKR appears to play a role in anisomycin-induced stress, we tested the exosomal samples for the presence of this enzyme as well (Figure 3b). While the level of PKR in all cell extracts did not significantly change upon anisomycin treatment (see Figure 1), both high and low anisomycin concentrations markedly increased its amounts in exosomes from wtPC12 cells, and high concentrations of anisomycin also triggered the cleavage of exosomal PKR. In sharp contrast, these changes did not happen in p143p53PC12 cells (Figure 3b).

## 4. Discussion

The study presented in this article is an extension of earlier work in this laboratory [11,36,38]: further details of anisomycin-induced stress and apoptotic signaling were analyzed in the PC12 model cell line. On one hand, some signaling effects of toxic and subinhibitory concentrations of anisomycin were compared; on the other hand, the role of the p53 tumor suppressor protein in these processes was studied using a PC12 subclone expressing a dominant-negative mutant p53 protein. Results of the 24-h time kinetic study indicated that (I) toxic anisomycin treatment (1 μg/mL) stimulated the cleavage of PKR and consequent phosphorylation of eIF2α, as well as proteolytic activation of caspase 3, 8 and 9 enzymes; low anisomycin concentration (10 ng/mL) had little impact on these processes in wtPC12cells; (II) the signaling events described above, were partially or completely blocked by the mutant p53 protein; (III) the p38 MAPK pathway was similarly stimulated by high and low anisomycin concentrations and was hardly affected by the functional state of the p53 protein; (IV) the JNK and ERK pathways displayed more complex alteration patterns (see below); (V) exosomal release of transmembrane TRAIL and soluble PKR levels were significantly increased by high and low anisomycin concentrations as well; these changes were strongly p53-dependent: exosome production was greatly reduced in p143p53PC12 cultures.

Ribotoxicity caused by anisomycin, as found earlier [36], was exemplified by strong activation of stress kinase (JNK and p38 MAPK) pathways, stimulation of the PKR/eIF2α axis, and activation of extrinsic (caspase-8-mediated) and intrinsic (caspase-9-dependent) apoptotic signaling, leading to strong stimulation of the executioner caspase-3 enzyme. These effects of anisomycin were strongly (PKR cleavage, eIF2α phosphorylation, proteolytic activation of caspases) or partially (JNK, p38 MAPK phosphorylation) p53-dependent.

The effect of low anisomycin concentration on stress and apoptotic signaling in wtPC12 cells is much weaker: JNK, p38 MAPK, and eIF2α phosphorylation are slightly increased without proteolytic activation of PKR. eIF2α phosphorylation may be caused by proteolytic cleavage/autophosphorylation-activated PKR or another eIF2α kinase (e.g., PERK, for a review see [8]). The mild cellular stress caused by non-toxic anisomycin concentration is followed by a weak apoptotic response: a weak and delayed proteolytic activation of all three caspases analyzed can be observed.

Most of the effects of low anisomycin concentration were reduced by the presence of the dominant inhibitory p53 protein (p38 MAPK and eIF2α phosphorylation, PKR cleavage, and proteolytic caspase activation could not be detected at all). A somewhat unexpected exception was the behavior of JNK enzymes. The level of JNK phosphorylation fluctuated during anisomycin treatments in both cell lines (in sharp contrast to the stable and sustained increase in p38 MAPK phosphorylation). One possible explanation for this phenomenon may be that the turnover of phosphates on JNK is much faster and the phosphorylation state of JNK is determined by the actual cellular conditions. Another remarkable observation is that the overall state of JNK phosphorylation is much higher in p143p53PC12 than in wtPC12 cells, and that low concentration anisomycin treatment strongly stimulated JNK phosphorylation (the kinetics of ERK phosphorylation was similar). A possible explanation for this observation is the presence of p53-regulated dual-specificity phosphatases (DUSPs) in these cells. The DUSP family of tyrosine phosphatases contains several enzymes that target members of the MAPK family (including JNKs, p38 MAPKs, and ERKs). Dephosphorylation of these protein kinases inactivates them, counteracting the stimulating effects of the dual specificity MAPK kinase enzymes (for reviews, see [39,40]). Several of the DUSP family members, including DUSP1 [41], DUSP2 [42], DUSP4 [43] and DUSP8 [39], are transcriptionally regulated by the p53 protein. Decreased expression of some of these protein phosphatases may account for the elevated phosphorylation level of JNK (and possibly ERK) enzymes in p143p53PC12 cells. It is also to be noted that this sustained JNK activation is not accompanied by proapoptotic signaling (caspase and PKR cleavage, eIF2α phosphorylation). Strong JNK phosphorylation is thus by itself not sufficient to induce apoptosis in PC12 cells, especially if it is accompanied by strong stimulation of the ERK pathway.

A possible explanation for caspase-8 activation upon anisomycin treatment is a paracrine-type stimulation of cultured cells by death ligands. Since we found intercellular signaling exosomes in stressed PC12 cultures [37], exosomal fractions from wtPC12 and p143p53PC12 cultures were analyzed for the presence of TRAIL, a powerful death ligand. The expression of the transmembrane 34 kDa isoform of TRAIL was strongly and similarly stimulated by high and low concentrations of anisomycin in wtPC12 cells. It thus appears that exosomal TRAIL signaling is unlikely to significantly contribute to the apoptotic effect of anisomycin: 10 ng/mL has a robust effect on exosomal TRAIL expression (Figure 3a) but caused a delayed, slight caspase-activation only (Figure 2). Other mechanisms may be responsible for caspase-8 activation upon toxic anisomycin treatment (e.g., increased secretion of soluble, non-exosomal TRAIL; a cross-talk between the intrinsic and extrinsic apoptosis pathway).

In addition to the transmembrane isoform of TRAIL, the PKR content of exosomes was similarly increased (Figure 3b). Neither intracellular TRAIL nor PKR levels were affected by anisomycin treatment (compare Figure 1b and Figure 3b), indicating the existence of a selective exosomal accumulation mechanism for these proteins. (The complex regulatory mechanisms governing exosome biogenesis and release, as well as cargo selection (as reviewed in [44]), are planned to be explored in future studies.) The exosomal changes were not detected in p143p53PC12 cells: TRAIL expression was almost completely absent, while changes in exosomal PKR expression were not observed. It can thus be concluded that the exosomal transfer of these stress-related proteins is completely under the control of the p53 protein, and as judged by the ERK loading control, even the production of exosomes is affected by the p53 protein. The cellular mechanism of these regulatory processes is at present unknown.

## 5. Conclusions

The PC12 rat pheochromocytoma cell line can be differentiated into neuronal cells resembling certain nerve cells of the central nervous system upon neurotrophin treatment. Such cells and their stable transfectants expressing exogenous proteins (like the cell line used in this study that expresses a dominant inhibitory p53 protein) provide useful cell culture models to study cellular processes that may take place in neurons of our nervous system. Besides neuronal differentiation, cell survival/death, and various forms of neuronal stress processes may also be studied in these cell lines. Taking these into consideration, observations obtained with these under mild and severe stress conditions may be cautiously extrapolated to signal transduction events during physiological and pathological processes in the brain. The present study was intended to further our understanding of signaling events caused by weak and strong stress stimulations and the role of the key apoptotic regulatory protein, p53, in the stress response.

## Figures and Tables

**Figure 1 biology-14-01634-f001:**
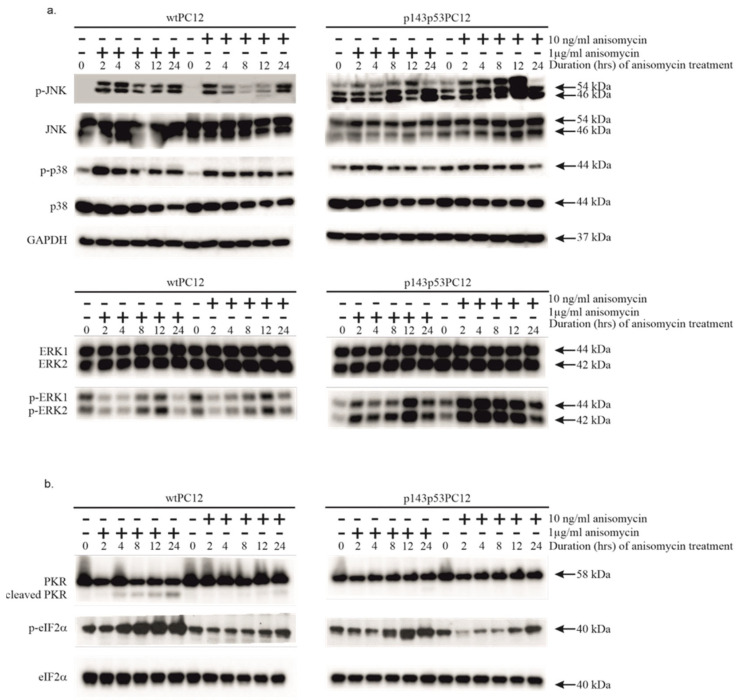
The effect of anisomycin treatment on various stress signaling proteins of wtPC12 and p143p53PC12 cells. Cell cultures were treated with high (1 µg/mL) or low (10 ng/mL) concentrations of anisomycin for the periods indicated in the headings of the figure. Cell extracts were subjected to Western-blot analysis as described in Materials and Methods using antibodies against proteins of the mitogen-activated protein kinase pathways (**a**) and the PKR-eIF2α pathway (**b**). Anti-GAPDH was used as a loading control. Densitometric quantification of Western blot bands is shown in the Appendix A.

**Figure 2 biology-14-01634-f002:**
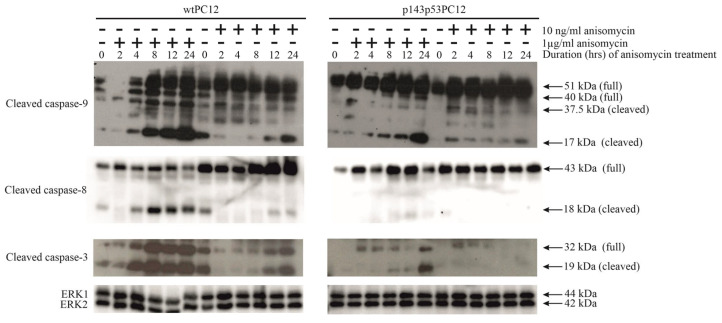
Anisomycin-induced caspase activation in wtPC12 and p143p53PC12 cells. The experiment was carried out as described in the legend to Figure 1. Anti-ERK1/ERK2 antibody was used as a loading control. Densitometric quantification of Western Blot bands is shown in the Appendix A.

**Figure 3 biology-14-01634-f003:**
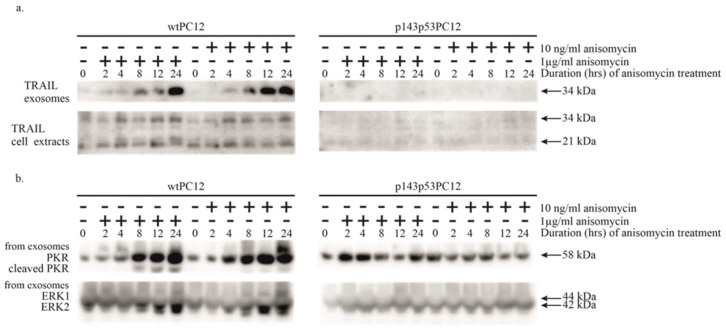
The effect of anisomycin treatment on exosomal TRAIL (**a**) and PKR (**b**) content. Cellular extracts were also tested for the TRAIL protein, treatment protocols were used as described in Materials and Methods and the legend to Figure 1. Anti-ERK1/ERK2 antibody was used as a loading control. Densitometric quantification of Western Blot bands is shown in the Appendix A.

## Data Availability

The original contributions presented in this study are included in the article/Appendix A. Further inquiries can be directed to the corresponding authors.

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
