# Peer review of "The Effect of a Dominant Inhibitory p53 Protein on Stress Responses Induced by Toxic and Non-Toxic Concentrations of Anisomycin in PC12 Cells"

_biology, 2025, doi:10.3390/biology14121634_

Round 1

Reviewer 1 Report

Comments and Suggestions for Authors

The manuscript titled “The effect of a dominant inhibitory p53 protein on stress responses induced by toxic and non-toxic concentrations of anisomycin in PC12 cells” by Schipp et al. introduces interesting concepts, such as the exosomal release of TRAIL and PKR and its dependence on p53, highlighting the role of the apoptotic regulatory protein p53 in the stress response. Nevertheless, some limitations in the experimental approach could affect the robustness of the findings. Below are some comments that may help improve the manuscript.

Comment 1:

Chemical formulas should include proper subscripts (e.g., H₂O, CO₂, Na₂HPO₄), and numerical values with exponents should use superscripts (e.g., 5 × 10⁶ instead of 5x10⁶) for clarity and adherence to standard scientific notation.

Comment 2:

In the presented gels, only a single endogenous control (GAPDH) is shown, and the band appears saturated, which may limit the accuracy of the comparative analysis of the investigated protein expression. I suggest performing densitometric quantification followed by appropriate statistical analysis to support and validate the claims reported in the manuscript.

Comment 3:

Replace “Intrestingly” with “Interestingly” (line 168).

Comment 4:

Could the authors clarify which kit was used for exosome isolation? When analyzing exosomes, it is important to include specific positive markers (e.g., CD9, CD63) to confirm the identity of the isolated fraction, as well as negative markers (e.g., calnexin, an endoplasmic reticulum marker, or GM130, a Golgi apparatus marker) to rule out potential contamination. We recommend confirming whether these validation steps were performed.

Author Response

Response to Reviewer #1:

We would like to thank Reviewer#1 for the thorough evaluation of our manuscript entitled “The effect of a dominant inhibitory p53 protein on stress responses induced by toxic and non-toxic concentrations of anisomycin in PC12 cells” and for the constructive comments that helped us improve the quality and clarity of the paper.

Please find our point-by-point responses below.

Comment 1: Chemical formulas should include proper subscripts (e.g., H₂O, CO₂, Na₂HPO₄), and numerical values with exponents should use superscripts (e.g., 5 × 10⁶ instead of 5x10⁶) for clarity and adherence to standard scientific notation.

Response 1: We thank the Reviewer for noting this. All chemical formulas and numerical values have been carefully checked and corrected to include proper subscripts and superscripts throughout the manuscript. For example, “CO2” has been replaced with “CO₂” (page 3, line 120), “Na2HPO4 × 7 H2O” with “Na₂HPO₄ × 7 H₂O” (page 4, lines 158, 159), and “5x10⁶” with “5 × 10⁶” (page 3, line 126, and page 4, line 150).

Comment 2: In the presented gels, only a single endogenous control (GAPDH) is shown, and the band appears saturated, which may limit the accuracy of the comparative analysis of the investigated protein expression. I suggest performing densitometric quantification followed by appropriate statistical analysis to support and validate the claims reported in the manuscript.

Response 2: We fully agree with the Reviewer that quantitative analysis would strengthen the conclusions. We quantified the Western blot band intensities using ImageJ software, normalized to the respective controls, and added the bar graphs to the Supplementary Materials (Figure S2).

Comment 3: Replace “Intrestingly” with “Interestingly” (line 168).

Response 3: We thank the Reviewer for pointing out this typographical error. The correction has been made in the revised version (page 4, line 177).

Comment 4: Could the authors clarify which kit was used for exosome isolation? When analyzing exosomes, it is important to include specific positive markers (e.g., CD9, CD63) to confirm the identity of the isolated fraction, as well as negative markers (e.g., calnexin, an endoplasmic reticulum marker, or GM130, a Golgi apparatus marker) to rule out potential contamination. We recommend confirming whether these validation steps were performed.

Response 4: We appreciate this valuable suggestion. Exosomes were isolated using the Total Exosome Isolation Reagent (Thermo Fisher Scientific, Cat. No. 4478359). We agree that the inclusion of exosomal and non-exosomal marker proteins (e.g., CD9, CD63, calnexin, GM130) would further confirm the identity and purity of the isolated vesicles. In the present study, our primary aim was to compare anisomycin-induced p53-dependent changes in the exosomal cargo, rather than to perform a comprehensive characterization of exosome markers. Therefore, we used a standardized and widely accepted precipitation-based isolation method, which reliably enriches small extracellular vesicles from cell culture media.

Reviewer 2 Report

Comments and Suggestions for Authors

Manuscript id: biology-3945588

Major Comments for Revision:

General comment: There are several typo errors in the manuscript, it is advised kindly check it. Also ensure the consistent formatting of units (µg/ml vs. ng/ml).

Comment 1: Clarify the "Non-Toxic" Concentration and its Mechanism.

Comment 2: Justify and explain the use of p53-Dominant Negative Model.

Comment 3:  How does the exosome carry TRAIL and PKR  fit into your explanation?

Comment 4: There are several gaps in the Introduction section that could cause readers to lose focus on the primary subject. Please clarify and make it easier to grasp.

Comment 5: It is advised to the authors to make graphical abstract and pathways of your findings in the results section, which could make this manuscript more interesting and easier to explain.

Comments on the Quality of English Language

The overall English may be improved because there are a few unclear passages and abrupt transitions between sections without any explanation.

Author Response

Response to Reviewer #2:

We would like to thank Reviewer#2 for the thorough evaluation of our manuscript entitled “The effect of a dominant inhibitory p53 protein on stress responses induced by toxic and non-toxic concentrations of anisomycin in PC12 cells” and for the constructive comments that helped us improve the quality and clarity of the paper. We also conducted a full language revision with native-level proofreading to improve clarity and coherence throughout the manuscript.

Please find our point-by-point responses below.

General comment: There are several typo errors in the manuscript, it is advised kindly check it. Also ensure the consistent formatting of units (µg/ml vs. ng/ml).

Response to general comment: The text had been thoroughly screened for typos, and corrections were made. As for the consistent formatting of units, we believe that our practice used in the original manuscript (e.g. 1 µg/ml, and 10 ng/ml of anisomycin) is also legitimate and would not confuse the reader. We therefore decided not to change the formatting. We hope that the reviewer can accept our decision.

Comment 1: Clarify the "Non-Toxic" Concentration and its Mechanism.

Response 1: The “non-toxic” concentration of anisomycin (10 ng/ml) used in this study does not inhibit protein synthesis and has no apoptotic effect in most cells. (In PC12 cells it shows a weak caspase-activating effect /see Fig.2/) The mechanism by which it causes the activation of stress kinase pathways is not known (as mentioned in the introduction).

Comment 2: Justify and explain the use of p53-Dominant Negative Model.

Response 2: Additional information describing the mechanism of action of the mutant p53 protein has been added to the Introduction, as requested (pages 2-3, lines 90-94).

Comment 3: How does the exosome carry TRAIL and PKR fit into your explanation?

Response 3: We also find this point an important issue of discussion. We tried to make a detailed interpretation of our exosomal results in the Discussion section. There are, however, several observations that at present can not be explained: (I) Are exosomal TRAIL and PKR involved in spreading and amplifying the apoptotic signal in wtPC12 cell populations? (II) If yes, why they do not cause apoptosis in cells treated with 10 ng/ml anisomycin? (III) What signaling mechanism may cross-talk with and strengthen the apoptotic effect of TRAIL/PKR? (IV) How is PKR selected for exosomal transport? At this moment we do not have the answers to these questions.

Comment 4: There are several gaps in the Introduction section that could cause readers to lose focus on the primary subject. Please clarify and make it easier to grasp.

Response 4: A few sentences have been added to the Introduction (highlighted in the text) to make the text clearer and smoother (page 2, lines 81-84).

Comment 5: It is advised to the authors to make graphical abstract and pathways of your findings in the results section, which could make this manuscript more interesting and easier to explain.

Response 5: We thank the Reviewer for this helpful suggestion. We agree that a graphical abstract or pathway illustration could further support the interpretation of our findings. However, we believe that the key mechanisms and relationships are already clearly described and visually represented in the existing figures and text.

Round 2

Reviewer 1 Report

Comments and Suggestions for Authors

Dear Authors,

Thank you for the revisions and for the work you have undertaken on the manuscript. I have carefully reviewed the changes and find them satisfactory. In light of the improvements made, I consider the manuscript suitable for publication and therefore recommend its acceptance.